# Transcatheter Aortic Valve Implantation: The New Challenges of Cardiac Rehabilitation

**DOI:** 10.3390/jcm10040810

**Published:** 2021-02-17

**Authors:** Simona Sperlongano, Francesca Renon, Maurizio Cappelli Bigazzi, Rossella Sperlongano, Giovanni Cimmino, Antonello D’Andrea, Paolo Golino

**Affiliations:** 1Department of Translational Medical Sciences, Division of Cardiology, University of Campania Luigi Vanvitelli, Monaldi Hospital, 80131 Naples, Italy; frances.renon@gmail.com (F.R.); mcappellibigazzi@gmail.com (M.C.B.); giovanni.cimmino@unicampania.it (G.C.); paolo.golino@unicampania.it (P.G.); 2Department of Experimental Sciences, University of Campania Luigi Vanvitelli, 80138 Naples, Italy; sperlonga1@libero.it; 3Department of Cardiology and Intensive Coronary Care, Umberto I Hospital, 84014 Nocera Inferiore, Italy; antonellodandrea@libero.it

**Keywords:** cardiac rehabilitation, aortic stenosis, transcatheter aortic valve implantation

## Abstract

Transcatheter aortic valve implantation (TAVI) is an increasingly widespread percutaneous intervention of aortic valve replacement (AVR). The target population for TAVI is mainly composed of elderly, frail patients with severe aortic stenosis (AS), multiple comorbidities, and high perioperative mortality risk for surgical AVR (sAVR). These vulnerable patients could benefit from cardiac rehabilitation (CR) programs after percutaneous intervention. To date, no major guidelines currently recommend CR after TAVI. However, emerging scientific evidence shows that CR in patients undergoing TAVI is safe, and improves exercise tolerance and quality of life. Moreover, preliminary data prove that a CR program after TAVI has the potential to reduce mortality during follow-up, even if randomized clinical trials are needed for confirmation. The present review article provides an overview of all scientific evidence concerning the potential beneficial effects of CR after TAVI, and suggests possible fields of research to improve cardiac care after TAVI.

## 1. Introduction

Transcatheter aortic valve implantation (TAVI) is a percutaneous intervention of aortic valve replacement (AVR), primarily indicated for elderly, vulnerable patients with severe aortic stenosis (AS). The benefit of cardiac rehabilitation (CR) in post-TAVI populations is still unclear, and CR is not currently recommended by the major guidelines for the management of patients after TAVI [1,2]. The present article provides an overview of all scientific evidence concerning the potential beneficial effects of CR, and the impact of CR programs on relevant clinical outcomes in post-TAVI populations.

## 2. Relevance of Cardiac Rehabilitation in Cardiovascular Disease

CR is a set of measures aimed at providing patients with cardiovascular disease with the optimum psychological and physical conditions to prevent the disease from progressing or potentially reversing its course [3]. The three main components of CR are exercise training, lifestyle modification, and psychological intervention (Figure 1). A CR program is cost-effective and it generally takes 3–4 weeks to complete. CR is strongly recommended in various cardiovascular diseases, such as in patients with stable angina pectoris, chronic heart failure, following myocardial infarction, surgical or percutaneous coronary revascularization, heart transplantation, and valvular surgery [4,5,6]. Among these subsets of patients, CR has been shown to improve exercise capacity, quality of life, and control of cardiovascular risk factors [7,8,9,10]. Moreover, CR reduces overall and cardiovascular mortality, as well as new hospital admissions after myocardial infarction [9].

In physiopathological terms, the benefits achieved by exercise-based CR can be attributed to various mechanisms: reduction in major risk factors, such as smoking, obesity, and arterial hypertension; improvement in lipid profile; improved medication adherence; depression identification and treatment; reduction in inflammation; ischemic preconditioning; improved endothelial function; improved insulin sensitivity and glucose levels; and more favorable fibrinolytic balance. CR also seems to favor a reverse cardiac remodeling with a reduction in left ventricular (LV) volumes, an increase in LV ejection fraction, and an improvement in LF and left atrial mechanics [11].

Although CR is strongly recommended after cardiac surgery, little is known about the impact of CR in patients with AS following TAVI.

## 3. Transcatheter Aortic Valve Implantation: Where the Need Arises for Cardiac Rehabilitation 

Degenerative AS is the most common valvular heart disease in developed countries, with a growing prevalence due to the ageing of the general population [1]. Its prognosis is poor when symptoms occur of dyspnea, angina, or syncope. Currently, there is no effective medical treatment for severe, symptomatic AS and AVR, and either surgery or transcatheter are the only possible therapeutic interventions.

Surgical AVR (sAVR) has been the gold standard treatment for a long time, but with an ageing and increasingly multimorbid population, the need for less invasive treatments has emerged. TAVI is recommended as the therapy of choice among inoperable patients by the current European guidelines on the management of valvular heart disease [1]. Moreover, it is an alternative to sAVR among patients at increased surgical risk, with the decision made by the heart team according to a patient’s clinical characteristics and anatomical and technical aspects. Therefore, to date, the target population for TAVI is composed of elderly, frail patients with multiple comorbidities and high perioperative mortality for sAVR. These subsets of patients may benefit from CR programs after percutaneous intervention, as although TAVI is less invasive than sAVR, it is still associated with a significant period of recovery.

Recently, the results of two randomized clinical trials (RCTs) comparing TAVI and sAVR (Placement of Aortic Transcatheter Valves [PARTNER] 3 and Evolut low risk) showed that the risk of death, disabling stroke, and rehospitalization for heart failure was lower in the TAVI group compared to the sAVR group, even among patients at low surgical risk [12,13]. In addition, an updated metanalysis, including seven RCTs and 8020 patients with severe symptomatic AS, reported a lower risk of all-cause mortality and stroke in TAVI than sAVR, irrespective of underlying surgical risk throughout 2 years of follow-up [14]. Thus, it is likely that TAVI indications will be extended to low-surgical-risk patients in the near future.

The increasing TAVI indications led to a growing interest in CR programs aimed at improving cardiac care after TAVI. Observational data from different geographical areas show variability in the participation to rehabilitation programs among post-TAVI patients [15,16].

## 4. Cardiac Rehabilitation after Transcatheter Aortic Valve Implantation: State of the Art and Evidence from the Literature

The Working Groups on Valvular Heart Disease, Thrombosis, and Cardiac Rehabilitation and Exercise Physiology of the European Society of Cardiology (ESC) recommend exercise-based CR after sAVR [17]. No recommendations are, on the contrary, provided for patients undergoing TAVI.

Over the years, various studies have investigated the safety and effectiveness of different CR programs in patients undergoing sAVR, whereas only a few studies have specifically addressed patients undergoing TAVI [18,19].

Apart from age and comorbidities, severe AS represents per se a contraindication to physical activity; consequently, patients usually undergo TAVI in a very deconditioned state. Moreover, hospital-acquired functional decline, defined as new or worsened functional decline during hospitalization, develops in at least 30% of hospitalized elderly patients. This functional decline has been observed to be independent of functional status and pre-operative frailty, and might be explained by a susceptibility to hospital-induced stresses. It is also independently associated with worse clinical outcomes [20]. As hospital-acquired functional decline is thought to be a result of physical inactivity during hospitalization, an acute phase of CR after TAVI might help in recovering patients’ functional performance. However, further research is needed to find the best strategy.

### 4.1. Safety

An exercise-based CR program after TAVI has been demonstrated to be safe without detrimental effects on prosthesis hemodynamics [18,19,21].

### 4.2. Exercise Tolerance

CR programs following TAVI show a gain in exercise tolerance, mainly assessed by the 6 min walking test distance (6MWD), and in some studies by the exercise test [18,22,23,24,25].

In their meta-analysis, Ribeiro et al. reported that the average increase in 6MWD was three times greater than the minimum clinically important difference in 6MWD observed in coronary artery disease patients who participated in a CR program after percutaneous coronary revascularization. The gain in exercise tolerance was similar after TAVI and SAVR [21].

In a study carried out by Tarro Genta et al., TAVI patients compared with sAVR patients exhibited a higher disability profile and a reduced exercise capacity, both in absolute terms and as improvement induced by CR [25]. TAVI patients showed a significantly shorter mean 6MWD at the beginning of CR program compared to sAVR patients in the study by Völler et al. [26]. Moreover, the change in 6MWD between admission and discharge from the CR program was significantly lower in the TAVI group compared to SAVR patients, remaining, however, significantly improved in both groups [26]. These results may be explained by patients undergoing TAVI being older and frailer than patients undergoing sAVR, and it is known that 6MWD is inversely related to age, and it is affected by comorbidities [27].

The effect of a structured exercise program versus usual care after TAVI was evaluated in a small randomized pilot trial [19]. Both groups underwent a standardized inpatient CR 2–3 weeks shortly after TAVI and before inclusion in the study, as part of local usual aftercare. The 8 week structured exercise program resulted in significant improvements in exercise capacity, assessed by a cardiopulmonary exercise test, and in muscular strength, compared to usual care. These results are, however, not generalizable to the general TAVI cohort, since only patients physically able to perform the exercise intervention were included.

### 4.3. Functional Independence and Quality of Life

An important goal of TAVI in elderly patients is to enhance their functional status. CR programs are associated with gains in functional independence, mainly assessed by the Bathel Index (BI), and in health-related quality of life [18,21]. Supervised training after TAVI contributes to lowering the risk of falls, which can favor home discharge and reduce the burden of bone fracture comorbidity [25].

It should not be forgotten that psychological endpoints, such as anxiety reduction, are part of quality of life, and they improve due to multicomponent rehabilitation programs [18,22].

### 4.4. Survival Benefit

Rehabilitation was shown to provide benefit in terms of 6 month survival after TAVI [28]. In particular, a CR program, enforcing physical exercise in addition to psychosocial training, was associated with a higher survival than a geriatric rehabilitation program. The advantage of survival provided by the rehabilitation program was predominantly driven by a reduction of non-cardiovascular mortality, which confirms previous observations [18,29]. No differences in valve hemodynamics and cardiac function were observed at 6 months after TAVI. Notably, this was an observational cohort study. Patients declining rehabilitation may have been behaviorally and socio-economically different from those who chose to perform rehabilitation. It was observed that patients declining rehabilitation were more likely to be depressed, of low socio-economic status, and physically inactive, predicting poorer clinical outcomes compared to patients receiving rehabilitation [28,30].

A trend toward better survival was also observed in the follow-up analysis of the randomized Safety, applicability and outcome of regular exercise training after transcatheter aortic valve implantation [SPORT: TAVI] pilot trial, with a non-significant difference presumably due to the small number of patients [31].

Though patients referred to rehabilitation are older and at higher risk [32], a recent observational study showed that 1 year outcomes after TAVI were not different between patients discharged home and patients discharged to rehabilitation facilities, whereas patients discharged to other institutions after the procedure showed higher rates of cardiac death, all-cause death, and bleeding [16]. Importantly, these are observational data, subject to selection bias, and with clinical and social variables influencing the mode of discharge. However, patients sent to rehabilitation showed similar long-term outcomes to that of the fitter patients sent home, and this may be linked to the effects of the rehabilitation program.

Confirmation of the effects of rehabilitation on hard outcomes is needed with further RCTs.

### 4.5. Cardiac Rehabilitation Derived Predictors of Outcome

Predictors of mid- and long-term adverse outcomes in patients who undergo TAVI are usually based on an assessment conducted before or at the time of percutaneous procedure. Some parameters deriving from CR programs can predict long-term outcomes. In particular, low exercise tolerance and severe residual disability at discharge from a residential CR program were shown to predict a 3 year follow-up [25]. However, patients of this study referred to rehabilitation were selected by their treating clinician (as is frequently the case with most of the studies conducted on rehabilitation after TAVI). Therefore, the conclusions may not be the same for more critical TAVI patients not referred to CR because of too-severe impairment or, conversely, for less critical patients directly discharged home after TAVI.

The lack of improvement in physical performance at 6 months after TAVI, evaluated through the 6MWD, was found to be an independent predictor of mortality and adverse cardiovascular outcomes during the ensuing 4 years [33]. CR programs, by improving functional status, may therefore favorably impact long-term clinical outcomes.

### 4.6. Long-Term Persistence of Cardiac Rehabilitation Effects

Intensified follow-up programs of multidisciplinary rehabilitation improve the clinical outcome of patients affected by cardiac disease, and should be offered whenever possible [3].

Acute exercise effects of rehabilitation after TAVI were mostly non-sustained over time in the follow-up of the SPORT: TAVI trial, indicating that once the intervention had stopped, patients were not willing or able to participate in regular home-based exercise [31]. In particular, 8 weeks of combined endurance and resistance exercise training shortly after TAVI produced long-term improvements in submaximal exercise capacity in oxygen uptake at anaerobic threshold (VO_2_AT), but not in oxygen uptake at peak workload (VO_2_peak), muscular strength, and quality of life compared to usual care. VO_2_AT is considered a more comprehensive marker of aerobic efficiency, while VO_2_peak describes the net limitation in exercise capacity. Both of them are strong predictors of all-cause mortality in heart failure patients. Although this has not specifically been investigated in TAVI populations, persisting improvements in submaximal exercise performance are thus very likely to facilitate activities of daily living, as these activities usually do not require maximal effort.

It is likely that adherence to regular exercise declines over time, resulting in a loss of initial clinical benefits. Strategies to promote the access to regular exercise programs may improve long-term outcomes after TAVI.

Figure 2 and Table 1 summarize the beneficial effects of CR following TAVI.

## 5. Center- vs. Home-Based Cardiac Rehabilitation

Traditionally, the offered CR programs are center-based, with activities performed in hospitals, gymnasiums, or sports centers. Home-based CR programs have been introduced to increase participation given their easier access and lower costs.

In patients with cardiac disease, home- and center-based CR are similarly effective in improving clinical status and health-related quality of life, and were shown to be safe [34]. Adherence to CR appears to be better with home-based programs, and this may favor long-term maintenance of rehabilitation benefits after the end of the programs [35,36].

Scarce evidence is presently available for home-based CR in high-risk populations, such is the majority of TAVI patients. It is reasonable to assume that even this subset of patients shows better adherence to home-based than center-based CR programs.

## 6. Prehabilitation and Clinical Optimization before TAVI

Evidence is accumulating that high-risk and frail cardiac patients might benefit from a preventive enrolment in rehabilitation strategies before cardiac and thoracic surgery: the prehabilitation (prehab) [37,38]. At present, there is no evidence on the optimal cardiac prehab program, although the postoperative outcome might be further improved by interventions targeting physical capacity, nutritional status, and psychological readiness to surgery [39]. Moreover, patients enrolled in prehab show increased compliance with post-procedural CR [37].

Since patients undergoing TAVI at present time are, by definition, at higher risk than patients undergoing surgery, the possible role of a cardiac prehab is of rising interest. Two RCTs are ongoing to evaluate the benefit of multicomponent cardiac prehab in patients at higher risk for clinical events before TAVI: The Protein and Exercise to Reverse Frailty in Older Men and women undergoing Transcatheter Aortic Valve Replacement (PERFORM-TAVR trial; NCT 03522454), and the Prehabilitation for Patients Undergoing Transcatheter Aortic Valve Replacement (TAVR-Prehab trial; NCT 03107897).

Identifying patients at increased risk of poor post-procedural outcome and adverse long-term prognosis is of pivotal importance to optimize their clinical status before TAVI (with the potential role of cardiac prehab), to minimize peri-procedural risks [40], and, potentially, to avoid futile procedures and address the patient to alternative therapeutic strategies or end-of-life care [41].

Frailty is a geriatric syndrome that diminishes the potential for functional recovery after TAVI. Different tools have been proposed to evaluate it, and the Essential Frailty Toolset (EFT) has been found to be the most robust predictor of outcomes, among other frailty scales, in older patients undergoing SAVR and TAVR [42,43]. In the same population, malnutrition was, independently of frailty status, a predictor of poor post-procedural outcome [44]. Screening for and correcting protein-energy malnutrition with pre-procedure nutritional supplementation might further improve post-TAVI outcomes [39].

On the spectrum of possibilities to improve clinical status before TAVI, a role for balloon aortic valvuloplasty (BAV) has been proposed. It could serve as a bridge-to-decision in patients with relative contraindications to TAVI, since improvement in mobility, New York heart association (NYHA) functional class, frailty, and left ventricular function have been described after BAV, allowing for definitive aortic treatment in up to 75% of high-risk cases [45,46,47].

## 7. Conclusions

CR is not currently recommended by the major American and European guidelines for patients undergoing TAVI. However, studies carried out over the last few years have proven that CR programs following TAVI are safe, improve patients’ exercise tolerance, functional independence, and quality of life. Moreover, preliminary data show a benefit on mid-term survival when CR is performed after TAVI, even if further and larger RCTs are needed for confirmation. Based on the existing literature, CR programs after TAVI (perhaps home-based), and intensified follow-up characterized by a multidisciplinary approach including medical care, exercise training, lifestyle advice, and psychological support, may be recommended.

## Figures and Tables

**Figure 1 jcm-10-00810-f001:**
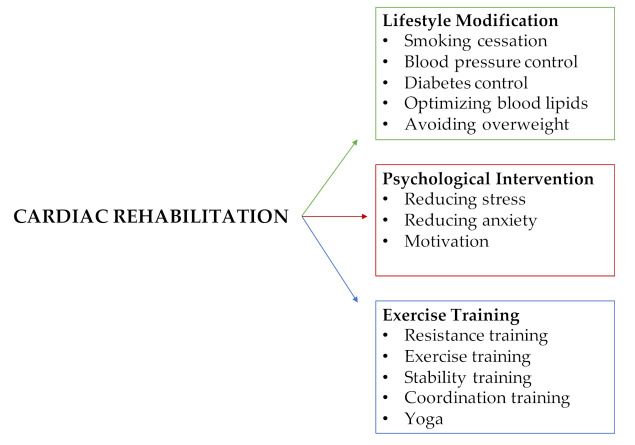
The main measures of cardiac rehabilitation: lifestyle modification, psychological intervention, and exercise training.

**Figure 2 jcm-10-00810-f002:**
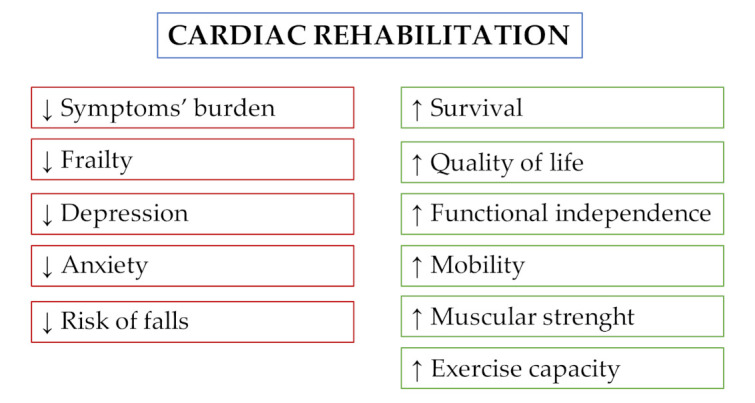
Effects of cardiac rehabilitation in patients undergoing transcatheter aortic valve implantation.

**Table 1 jcm-10-00810-t001:** Overview of the main clinical studies on cardiac rehabilitation after transcatheter aortic valve implantation.

Year	Study	No. Patients	Type of Study	Components of Cardiac Rehabilitation Program	Measures	Results
2014	Zanettini et al. [18]	60	Prospective observational single-arm	- Optimization of drug therapy - Nutritional intervention - Functional recovery and disability treatment (bed exercises, sitting calisthenics, ambulatory training, aerobic training with cicloergometer or treadmill, and calisthenics)	- Safety (clinical and echocardiographic parameters) - Exercise tolerance (6MWT) - Functional independence (modified BI) - Health-related QoL (EQ-VAS)	- Excellent PV performance - Improved functional capacity - Improved autonomy - Improved QoL
2016	Pressler et al. [19]	27	Randomized controlled pilot trial	Endurance and resistance exercise	- Safety (clinical and echocardiographic parameters) - Functional capacity (VO2peak at CPET) - Muscular strength (1-RM on 5 machines) - Exercise tolerance (6MWT) - QoL (KCCQ, SF-12) - NT-proBNP	- Safety of RP - Improved exercise capacity - Improved muscular strength - Improved QoL relative to physical function - Decreased symptom burden
2017	Ribeiro et al. [21]	292 TAVI patients	Meta-analysis	Gymnastic, aerobic exercise (cycling or treadmill), respiratory workout, calisthenics, resistance training, ambulatory training, bed and sitting exercises	- Safety (clinical and echocardiographic parameters) - Functional capacity (CPET) - Exercise tolerance (6MWT) - Functional independence (BI, FIM) - Health-related QoL (HADS, EQ-VAS) - All cause and CV mortality	- Safety of RP - Improved exercise capacity - Improved functional independence - Improved health-related QoL
2017	Eichler et al. [22]	136	Prospective cohort study	- Patient education - Diet counselling - Psychological support - Risk factors management - Training (bicycle, walking, and strength training)	- Exercise tolerance (6MWT) - QoL (SF-12, HADS) - Frailty-Index	- Improved exercise capacity - Improved QoL - Reduced anxiety - Reduced frailty
2014	Russo et al. [23]	78 TAVI patients	Prospective observational study	Low/medium intensity exercise protocol: respiratory workout, aerobic session (cycling), and callisthenic exercise	- Functional capacity (6MWT, CPET) - Functional independence (BI)	- Safety of RP - Improved independence - Improved mobility - Improved functional capacity
2014	Fauchère et al. [24]	34 TAVI patients	Retrospective observational study	Low/medium intensity exercise protocol: gymnastic, aerobic exercise, and respiratory workout sessions	- Functional independence (FIM) - Psychological distress (HADS) - Exercise tolerance (6MWT)	- Improved exercise capacity - Improved functional independence
2017	Tarro Genta et al. [25]	65	Prospective observational study	Aerobic incremental exercise training program: sessions of cycling or treadmill, and respiratory training	- Safety (clinical and echocardiographic parameters) - Functional capacity (6MWT) - Comorbidity (CIRS-CI) - Disability (BI) - Risk of falls (MFS)	- Safety of RP - Improved disability - Improved functional capacity - Reduced risk of falls
2014	Völler et al. [26]	76 TAVI patients	Observational study and propensity score analysis	- Patient education - Psychological training (including stress management, Tai Chi, and progressive muscle relaxation) - Aerobic training (bicycle ergometer) outdoor walking, gymnastics, and resistance training of the lower extremities	- Safety (clinical and echocardiographic parameters) - Functional capacity (6MWT, cycle-exercise test) - Emotional status (HADS)	- Safety of RP - Improved physical performance - Trend toward reduced depressive symptoms
2018	Butter et al. [28]	1017	Longitudinal cohort study	- Patient health education - Lifestyle and dietary advice - Psychological support - Physical activity (aerobic and resistance training)	- 6 months mortality - Cardiac function (LVEF, peak valve gradient, aortic insufficiency, NYHA functional class, NT-proBNP)	Higher survival at 6 months in CR patients (reduction in non-CV mortality)
2018	Pressler et al. [31]	17	Randomized controlled pilot trial	Endurance and resistance training	- Functional capacity (VO2peak and VO2AT at CPET) - Muscular strength (1-RM) - Exercise tolerance (6MWT) - QoL (KCCQ, SF-12) - Symptoms (NYHA functional class) - NT-proBNP - Echocardiographic parameters	- Long-term improvement in submaximal exercise capacity (VO2AT) - Trend toward improved survival - Not long-term persistence of improvement in VO2peak, muscular strength and QoL

BI: Barthel index; CIRS-CI: cumulative illness rated state-comorbidity index; CPET: cardiopulmonary exercise test; CR: cardiac rehabilitation; CV: cardiovascular; EQ-VAS, EuroQol: Questionnaire visual analogue scale; FIM: functional independence measure; HADS: hospital anxiety and depression scale; KCCQ: Kansas City cardiomyopathy questionnaire; LVEF: left ventricular ejection fraction; MFS: Morse fall scale; NT-proBNP: N-terminal pro-brain natriuretic peptide; NYHA: New York heart association; PV: prosthetic valve; QoL: quality of life; RP: rehabilitation program; SF-12: medical outcomes study 12-item short-form health survey; TAVI: transcatheter aortic valve implantation; VO2AT: oxygen uptake at the anaerobic threshold; VO2peak: peak oxygen uptake; 1-RM: 1-repetition maximum; 6MWT: 6 min walking test.

## Data Availability

Not applicable.

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
