# Peer review of "Transcatheter Aortic Valve Implantation: The New Challenges of Cardiac Rehabilitation"

_jcm, 2021, doi:10.3390/jcm10040810_

Round 1
Reviewer 1 Report
In the present paper, Sperlongano et al report a review about the advantages of cardiac rehabilitation (CR) after transcatheter aortic valve implantation (TAVI). Literature about this issue is summarized and possible benefits of CR in this setting are underscored.
An increasing number of patients all around the world are now candidates for TAVI, however CR is not indicated according to current guidelines; so the issue reported in the paper is relevant and worthy of attention.
The review is complete and well done
I have some comments
Paragraph 2.
Page 1 line 39 change “measure” with “component”
Page 2 line 45-50: this part of the text deserve some corrections:
It should be specified that CR improves functional capacity and quality of life in all the conditions reported, but the reduction of mortality has been proven only in patients with a recent myocardial infarction. In other diseases, there is not clear proof of reduction of mortality.
In physio pathological terms, is universally accepted that most of the benefits of exercise training are due to peripheral adaptions, while its effect on cardiac remodelling is often negligible.
Paragraph 4
Authors should specify the core components of the rehabilitation programmes of the analysed studies.
Because many of the patients treated by tavi are very old, probably aerobic exercise was not indicated for all of them.
Authors could add a column in table 1 reporting the core components of CR n each study
Author Response
We thank the reviewer for his/her comments to our manuscript and we invite him/her to see the attachment with our response.

Reviewer 2 Report
This is a very timely and well written overview about the role of cardiac rehab in patients undergoing TAVI
I would suggest to further stress three points:
-Afilalo clearly showed that EFT score can stratify patients with poor outcome after TAVI. In addition, Afilalo showed that malnutrition is a strong determinant of poor prognosis. Probably, a CR program after TAVI cannot change this outcome because the large majority of events is soon after TAVI. Can the Authors speculate about the possibility to introduce CR before TAVI? To improve patient's condition and increase the benefit of TAVR?
-Some Authors suggest the role of balloon aortic valvuloplasty as bridge to improve patient's condition before TAVI (if their baseline condition are poor). Can the Author discuss this point?
-The Authors suggest the need of CR facilities. I do not agree. These facilities fail after MI and recently also after cardiac surgery due to high costs. I believe that outpatient service and mixed model based on supervised sessions and home-based exercise program can be more safe, effective and sustainable (see similar findings in MI patients PMID: 32144189 and PMID: 31806139)
Author Response

(The authors gave the same response as above.)

Round 2
Reviewer 1 Report
Authors modified the paper according to my comments.
I don't have any more suggestions.